# INFORMATION CONDENSING ACTIVE LEARNING

## ABSTRACT

We introduce Information Condensing Active Learning (ICAL), a batch mode model agnostic Active Learning (AL) method targeted at Deep Bayesian Active Learning that focuses on acquiring labels for points which have as much information as possible about the still unacquired points. ICAL uses the Hilbert Schmidt Independence Criterion (HSIC) to measure the strength of the dependency between a candidate batch of points and the unlabeled set. We develop key optimizations that allow us to scale our method to large unlabeled sets. We show significant improvements in terms of model accuracy and negative log likelihood (NLL) on several image datasets compared to state of the art batch mode AL methods for deep learning.

## 1 INTRODUCTION

Machine learning models are widely used for a vast array of real world problems. They have been applied successfully in a variety of areas including biology (Ching et al., 2018), chemistry (Sanchez-Lengeling and Aspuru-Guzik, 2018), physics (Guest et al., 2018), and materials engineering (Aspuru-Guzik and Persson, 2018). Key to the success of modern machine learning methods is access to high quality data for training the model. However such data can be expensive to collect for many problems. Active learning (Settles, 2009) is a popular methodology to intelligently select the fewest new data points to be labeled while not sacrificing model accuracy. The usual active learning setting is *pool-based* active learning where one has access to a large unlabeled dataset $\mathcal{D}_U$ and uses active learning to iteratively select new points from $\mathcal{D}_U$ to label. Our goal in this paper is to develop an active learning acquisition function to select points that maximize the eventual test accuracy which is also one of the most popular criteria used to evaluate an active learning acquisition function.

In active learning, an *acquisition function* is used to select which new points to label. A large number of acquisition functions have been developed over the years, mostly for classification (Settles, 2009). Acquisition functions use model predictions or point locations (in input feature or learned representation space) to decide which points would be most helpful to label to improve *model accuracy*. We then query for the labels of those points and add them to the training set. While the past focus for acquisition functions has been the acquisition of one point at a time, each round of label acquisition and retraining of the ML model, particularly in the case of deep neural networks, can be expensive. Furthermore in several applications like biology, it can be much faster to do acquisition of a fixed number of points in parallel versus sequentially. There have been several papers, particularly in the past few years, that try to avoid this issue by acquiring points in *batch*. As our goal is to apply AL in the context of modern ML models and data, we focus in this paper on batch-mode AL.

Acquisition functions can be broadly thought of as belonging to two categories. The ones from the first category directly focus on minimizing the error rate post-acquisition. A natural choice of such an acquisition function might be to acquire labels for points with the highest uncertainty or points closest to the decision boundary (Uncertainty sampling can be directly linked to minimizing error rate in the context of active learning Mussmann and Liang (2018)). In the other category, the goal is to get as close as possible to the true underlying model. Thus here, acquisition functions select points which give the most amount of knowledge regarding a model's parameters where knowledge is defined as the statistical dependency between the parameters of the model and the predictions for the selected points. Mutual information (MI) is the usual choice for the dependency, though other choices are possible. For well-specified model spaces, e.g. in physics, such a strategy can identify the correct model. In machine learning, however, models are usually mis-specified, and thus the

metric of evaluation even for model-identification acquisition functions is how successful they are at reducing test error. Given this reality, we follow the viewpoint of trying to *minimize the error rate* of the model post-acquisition.

Our strategy is to select points that we expect would provide substantial information about the labels of the rest of the unlabeled set, thus reducing model uncertainty. We propose acquiring a batch of points $\mathcal{B}$ such that the model's *predictions* on $\mathcal{B}$ have as high a statistical dependency as possible with the model's *predictions* on the entire unlabeled set $\mathcal{D}_U$. Thus we want a batch $\mathcal{B}$ that *condenses* the most amount of information about the model's predictions on $\mathcal{D}_U$. We call our method Information Condensing Active Learning  (ICAL).

A key desideratum for our acquisition function is to be model agnostic. This is partly because the model distribution can be very heterogeneous. For example, ensembles which are often used as a model distribution can consist of just decision trees in a random forest or different architectures for a neural network. This means we cannot assume any closed form for the model's predictive distribution, and have to resort to Monte Carlo sampling of the predictions from the model to estimate the dependency between the model's predictions on the query batch and the unlabeled set. MI, however, is known to be hard to approximate using just samples (Song and Ermon, 2019). Thus to scale the method to larger batch sizes, we use the Hilbert-Schmidt Independence Criterion (HSIC), one of the most powerful extant statistical dependency measures for high dimensional settings. Another advantage of HSIC is that it is *differentiable*, which as we will discuss later, can allow applications of the acquisition function to areas where MI would be difficult to make work.

To summarize, we introduce Information Condensing Active Learning  (ICAL) which maximizes the amount of information being gained with respect to the model's predictions on the unlabeled set of points. ICAL is a batch mode acquisition function that is model agnostic and can be applied to both classification and regression tasks. We then develop an algorithm that can scale ICAL to large batch sizes when using HSIC as the dependency measure between random variables. As our method only needs samples from the posterior predictive distribution which can be obtained for both regression and classification tasks, it is applicable to both.

## 2   Related work

A review of work on acquisition functions for active learning prior to the recent focus on deep learning is given by Settles (2009). The BALD (Bayesian Active Learning by Disagreement) (Houlsby et al., 2011) acquisition function chooses a query point which has the highest mutual information about the model parameters. This turns out to be the point on which individual models sampled from the model distribution are confident about in their prediction but the overall predictive distribution for that point has high entropy. In other words this is the point on which the models are individually confident but disagree on the most.

In Guo and Schuurmans (2008) which builds on Guo and Greiner (2007), they formulate the problem as an integer program where they select a batch such that the post acquisition model is highly confident on the training set and has low uncertainty on the unlabeled set. While the latter aspect is related to what we do, they need to retrain their model for every candidate batch they search over in the course of trying to find the optimal batch. As the total number of possible batches is exponential in the size of the unlabeled set, this can get too computationally expensive for neural networks limiting the applicability of this approach. Thus as far as we know, Guo and Schuurmans (2008) has only been applied to logistic regression. BMDR (Wang and Ye, 2015) queries points that are as close to the classifier decision boundary as possible while still being representative of the overall sample distribution. The representativeness is measured using the maximum mean discrepancy (MMD) (Gretton et al., 2012) of the input features between the query batch and the set of all points, with a lower MMD indicating a more representative query batch. However this approach is limited to classification problems, as it is based on a decision boundary. BMAL (Hoi et al., 2006) selects a batch such that the Fisher information matrices for the total unlabeled set and the selected batch are as close as possible. The Fisher information matrix is however quadratic in the number of parameters and thus infeasible to compute for modern deep neural networks. FASS (Filtered Active Subset Selection) (Wei et al., 2015) picks the most uncertain points and then selects a subset of those points

that are as similar as possible to the whole candidate batch which favors points that can represent the diversity of the initial set of most uncertain points.

Recently active learning methods have been extended to the deep learning setting. Gal et al. (2017) adapts BALD (Houlsby et al., 2011) to the deep learning setting by using Monte Carlo Dropout (Gal and Ghahramani, 2016) to do inference for their Bayesian Neural Network. They extend BALD to the batch setting for neural networks with BatchBALD (Kirsch et al., 2019). In Pinsler et al. (2019), they adapt the Bayesian Coreset (Campbell and Broderick, 2018) approach for active learning, though their approach requires a batch size that changes for every acquisition. As the neural network decision boundary is intractable, DeepFool (Ducoffe and Precioso, 2018) uses the concept of adversarial examples (Goodfellow et al., 2014) to find points close to the decision boundary. However this approach is again limited to classification tasks. FF-Comp (Geifman and El-Yaniv, 2017), DAL (Gissin and Shalev-Shwartz, 2019), Sener and Savarese (2017), and BADGE (Ash et al., 2019) operate on the learned representation, as that is the only way the methods incorporate feedback from the training labels into the active learning acquisition function, and they are thus not model-agnostic, as they are not extendable to any model distribution where it is difficult to have a notion of a common representation – as in a random forests or ensembles, etc. where the learned representation is a distribution and not a single point. This is also the case with the model distribution – MC-dropout – we use in this paper.

There is also extensive prior work on exploiting Gaussian Processes (GPs) for Active Learning (Houlsby et al., 2011; Krause et al., 2008). However GPs are hard to scale especially for modern image datasets.

# 3 Background

**Statistical background** The entropy of a distribution is defined as $H(Y) = -\sum_{x \in \mathcal{X}} p_x \log(p_x)$, where $p_x$ is the probability of the $x$. Mutual information (MI) between two random variables is defined as $\mathbb{I}[X;Y] = \sum_{x \in \mathcal{X}} \sum_{y \in \mathcal{Y}} p(x,y) \log(\frac{p(x,y)}{p(x)p(y)})$, where $p(x,y)$ is the joint probability of $x, y$. Note that $\mathbb{I}[X;Y] = H(Y) - H(Y|X) = H(X) - H(X|Y)$. By posterior predictive distribution $y_x$ we mean $\int_\theta p(y|x, \theta) p(\theta|\mathcal{D}) d\theta$ where $y$ is the prediction, $x$ the input point, $\theta$ the model parameters, and $\mathcal{D}$ the training data. $\mathcal{M}$ is the distribution of models (parametrized by $\theta$) we wish to choose from via active learning. As mentioned before, we use MC-dropout for our model distribution by sampling random dropout masks and use the same set of dropout masks across points to generate joint predictions.

**Hilbert-Schmidt Independence Criterion (HSIC)** Suppose we have two (possibly multivariate) distributions $\mathcal{X}, \mathcal{Y}$ and we want to measure the dependence between them. A well known way to measure it is using *distance covariance* which intuitively, measures the covariance between the *distances* of pairs of samples from the joint distribution $P_{XY}$ and the product of marginal distributions $P(X), P(Y)$ (Székely et al., 2007). HSIC can simply be thought of as distance covariance except in a kernel space (Sejdinovic et al., 2013b). A (sample) kernel matrix $k^X$ is a matrix whose $ij$th element is $k(x_i, x_j)$ where $k$ is the kernel function and $x_i, x_j$ are the $i, j$th samples from $X$. Further details are in the Appendix.

**Acquisition Function** Let the batch to acquire be denoted by $\mathcal{B}$ with $B = |\mathcal{B}|$. Given a model distribution $\mathcal{M}$, training data $\mathcal{D}_{train}$, unlabeled data $\mathcal{D}_U$, input space $\mathcal{X}$, set of labels $\mathcal{Y}$ and an acquisition function $\alpha(x, \mathcal{M})$, we decide which batch of points to query next via:

$$\mathcal{B}^* = \arg\max_{\mathcal{B}} \alpha(\mathcal{B}, \mathcal{M})$$

# 4 Motivation

As mentioned previously, our goal is to acquire points that will give us as much information as possible about the still-unlabeled points, thereby increasing the confidence of the model's predictions.

As we will demonstrate shortly, there are situations where modern active learning methods do not select the points that optimally decrease the uncertainty of prediction on the unlabeled data. More formally, the examples below show that the choice of $x \in U$ that optimizes oft-used acquisition functions may not be optimal for decreasing the entropy of predictions $(\sum_{x' \in U, x' \neq x} H(y_{x'}))$ over the remaining points post-acquisition. If we wish to optimize test-set accuracy, this can be problematic: for well-calibrated models, we should expect worse average entropy (uncertainty) to roughly correspond to an increase in the number of errors. This is similar to cross entropy loss being a good proxy for 0-1 loss. Below we illustrate our points with two examples and from results on EMNIST.

**Example 1**   Suppose we have an image dataset which is highly imbalanced with 90% cars, 9% planes, and 1% ships. Then a small increase in accuracy for the car category would lead to a much larger reduction in the overall error rate versus a large increase in accuracy for the ships category. However, given the dominance of the cars category in the loss, the uncertainty of prediction on the ships category is likely to be much higher. Thus the max-entropy criterion is more likely to choose points from the pool set that turn out to be ships.

**Example 2**   Similar to the previous example, here we demonstrate that picking the point with the most amount of information with respect to the model parameters is not optimal for decreasing the prediction uncertainty on the still unlabeled data. The main idea behind this example is that if you have points which form a non-trivial fraction of the dataset and have a lot of correlation between their predictive distributions, then while any of the points may not give a lot of information about which underlying model is the best one, getting the labels for one of the points will greatly reduce the predictive uncertainty for the labels for the other points given the predictive distribution correlation. As these points are a non-trivial fraction of the dataset, reducing the predictive uncertainty on them will have a big impact on the error rate. The example in the Appendix formalizes this intuition.

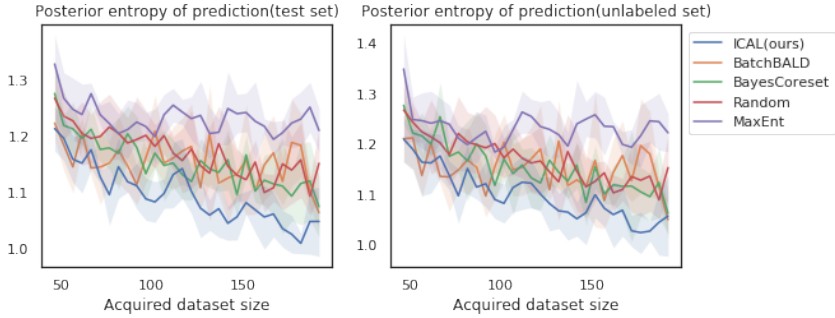

Figure 1: Mean posterior entropy of the predictions after each acquisition on EMNIST.

These observations motivate our formulation of the Information Condensing Active Learning (ICAL) acquisition function that selects the set of points whose acquisition would maximize the information gained about the predictive distribution on the unlabeled set. As posterior prediction entropy should be minimized by maximizing Mutual Information (MI) between predictions for unlabeled points and prediction for selected points, ideally ICAL would use MI or related criteria to select points.

**EMNIST results**   In Figure 1, we show the average posterior entropy of the model's predictions for our method compared to BatchBALD, BayesCoreset, and Random acquisition. As can be seen from the figure, ICAL reduces the average posterior entropy much more effectively compared to the other two. Details of this experiment are in Section 6.2.

# 5   Information Condensing Active Learning (ICAL)

In this section we present our acquisition function. As before, let $\mathcal{D}_{train}$ be the training points, $\mathcal{D}_U$ the unlabeled points, $y_x$ the random variable denoting the prediction for $x$ by the model trained on $\mathcal{D}_{train}$, and $\mathfrak{d}$ the dependency measure being used. Then

$$\alpha_{ICAL}(\{x_1, \ldots, x_B\}, \mathfrak{d}) = \frac{1}{|\mathcal{D}_U|} \sum_{x' \in \mathcal{D}_U} \mathfrak{d}(y_{x'}, \{y_{x_1}, \ldots, y_{x_B}\})$$

that is, we try to find the batch that has highest average dependency with respect to the unlabeled points' marginal predictive distribution.

## Scaling $\alpha_{ICAL}$ estimation

As we mentioned in the introduction, we can use MI as the dependency measure $\mathfrak{d}$ but it is tricky to estimate MI using just samples from the distribution, particularly high-dimensional or continuous variables. Furthermore, MI estimators are usually not differentiable. Thus if we wanted to apply ICAL to domains where the pool set is continuous and infinite (for example, if we wanted to query gene expression perturbations for a cell), we would run into obstacles. This motivates our choice of $HSIC$ as the dependency measure. In addition to being differentiable, $HSIC$ has better empirical sample complexity for measuring dependency as opposed to estimators for MI. Indeed, popular MI estimators have been found to have variance with respect to ground truth MI that increases exponentially with the MI value Song and Ermon (2019). $HSIC$ has also been successfully used in the related context of feature selection via dependency maximization in the past Da Veiga (2015); Song et al. (2012). Furthermore, $HSIC$ is the Maximum Mean Discrepancy (MMD) between the joint distribution and the production of marginals. MMD is known to be $\leqslant \frac{1}{2}$ KL-divergence Ramdas et al. (2015) and thus $HSIC \leqslant \frac{1}{2}$ MI. Thus we use $HSIC$ as the dependency measure for the rest of the paper.

Naively implementing $\alpha_{ICAL}(\mathcal{B}, HSIC)$ would require $O(|\mathcal{D}_U|m^2 B \cdot C)$ steps per candidate batch being evaluated where $C$ is the number of classes, $m$ is the number of samples taken from $p(y_{1:B})$ ($O(m^2 B)$ to estimate $HSIC$ which we need to do $|\mathcal{D}_U|$ times).

However, recall that $HSIC$ is a function of solely the kernel matrices $k^x$ corresponding to the random variables (Appendix) – in this case $y_x, x \in \mathcal{D}_U$. Now one can define the matrix $k^* = \frac{1}{|\mathcal{D}_U|} \sum_{x \in \mathcal{D}_U} k^x$. We can then prove the following propositions (proofs are in the Appendix).

**Proposition 1** $k^*$ is a valid kernel matrix.

**Proposition 2** $\sum_{x' \in \mathcal{D}_U} \widehat{HSIC}(k^{x'}, k^{x \in \mathcal{B}}) = \widehat{HSIC}(\sum_{x \in \mathcal{D}_U} k^x, k^{x \in \mathcal{B}})$

where $k^{x \in \mathcal{B}} = k^{x_1}, \ldots, k^{x_B}, x_i \in \mathcal{B}$ and $\widehat{HSIC}$ denotes the sample for $HSIC$. Using this reformulation, we only have to compute $k^* = \frac{1}{|\mathcal{D}_U|} \sum_{x \in \mathcal{D}_U} k^x$ once per acquisition round. This lowers the computation cost to $O(|\mathcal{D}_U|m^2 \cdot C + m^2 B \cdot C)$. Estimating $HSIC$ would still require $m$ to increase very rapidly with $B$ (proportional to the dimension of the joint distribution). To get around this but still maintain batch diversity, we try two strategies.

For regular ICAL, we average the kernel matrices of points in the candidate batch. We then subsample $r$ points from $\mathcal{D}_U$ every time a point is added to the batch and only compare the dependency with those. This effectively introduces *noise* in the $HSIC$ estimation. We find in practice, that this is sufficient to acquire a diverse batch, as evidenced by Figure 3. This seems to be the case even for very large batches, and has the added benefit of further lowering the computational cost for evaluating a candidate batch to $O(rm^2 \cdot C + 2 \cdot m^2 \cdot C)$. We use $r = 200$ for all our experiments.

We develop another strategy we call ICAL-pointwise which computes the marginal increase in dependence as a result of adding a point to the batch. If a point is highly correlated with elements of the current batch, the marginal increase would be negligible, making the point much less likely to be selected. The two variants perform very similarly despite ICAL-pointwise's slight advantage in the early acquisitions. ICAL-pointwise however requires much less time for equivalent performance which we discuss briefly in Section 5.2 and more fully in the Appendix. For ease of presentation, we use ICAL in the Results section and defer the full description and evaluation of ICAL-pointwise to the Appendix.

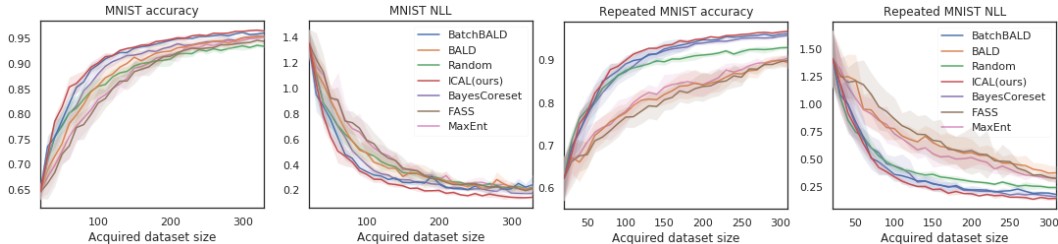

Figure 2: Performance on MNIST and repeated-MNIST. Accuracy and NLL after each acquisition.

As there are an exponential number of candidate batches, an exhaustive search to find the optimal batch is infeasible. For ICAL we use a greedy forward selection strategy to build the batch and find that it performs well empirically. As the $\arg\max$ over all of $\mathcal{D}_U$ has to be computed every time a new point is being selected for the batch, and we have to perform this operation for each point that is added to the batch, this gives a computation cost of $O((r^2 m^2 + |\mathcal{D}_U| m^2 B + m^2 B) \cdot C) = O(|\mathcal{D}_U| m^2 B \cdot C)$. It is possible that global nonlinear optimization of the batch ICAL criterion would work even better than greedy optimization already does with respect to state of the art methods. Efficient techniques for doing this optimization are not obvious and beyond the scope of this work. Even if we used gradient based techniques to construct the batch, gradient based optimization for nonlinear problems usually only leads to local and not global optima. We note however that greedy forward selection is a popular technique that has been successfully used in a large variety of contexts (Da Veiga, 2015; Blanchet et al., 2008). Optimizations to scale ICAL even further as well as the full Algorithm are detailed in the Appendix.

# 6    Results

We demonstrate the effectiveness of ICAL using standard image datasets including MNIST (LeCun et al., 1998), Repeated MNIST (Kirsch et al., 2019), Extended MNIST (EMNIST) (Cohen et al., 2017), fashion-MNIST, and CIFAR-10 (Krizhevsky et al., 2009). We compare ICAL with three state of the art methods for batched active learning acquisition – BatchBALD, FASS, and BayesCoreset. We also compare against BALD and Max Entropy (MaxEnt) which are not explicitly designed for batched selection, as well as against a Random acquisition baseline. Details of the acquisition functions are in the Appendix. ICAL consistently outperforms BatchBALD, FASS, and BayesCoreset on accuracy and negative log likelihood (NLL).

Throughout our experiments, for each dataset we hold out a fixed test set for evaluating model performance after training and a fixed validation set for training purposes. We retrain the model from the beginning after each acquisition to avoid correlation of subsequently trained models, and we use early stopping after 3 (6 for ResNet18) consecutive epochs of validation accuracy drop. Following (Gal et al., 2017), we use Neural Networks with MC dropout (Gal and Ghahramani, 2016) as a variational approximation for Bayesian Neural Networks. We simply use a mixture of rational quadratic kernels for $HSIC$, which has been used successfully with kernel based statistical dependency measures in the past, with mixture length scales of $\{0.2, 0.5, 1, 2, 5\}$ as in (Bińkowski et al., 2018). All models are optimized with the Adam optimizer (Kingma and Ba, 2014) using learning rate of 0.001 and betas (0.9,0.999). The small batch size experiments are repeated 6 times with different seeds and a different initial training set for each run, with balanced label distribution across all classes. The same set of seeds is used for different methods on the same task. 8 different seeds are used for large batch size experiments using CIFAR datasets.

## 6.1    MNIST and Repeated MNIST

We first examine ICAL's performance on MNIST, which is a standard image dataset for handwritten digits. We further test out the scenario where duplicated data points exist (repeated MNIST) as proposed in (Kirsch et al., 2019). Each data point in MNIST is replicated three times in repeated-

MNIST, and isotropic Gaussian noise with std=0.1 is added after normalizing the image. We use a CNN consists of two convolutional layers with 32 and 64 5x5 convolution filters, each followed by MC dropout, max-pooling and ReLU. One fully connected layer with 128 hidden units and MC dropout is used after convolutional layers and the output soft-max layer has dimension of 10. All dropout uses probability of 0.5, and the architecture achieved over 99% accuracy on full MNIST. We use a validation set of size 1024 for MNIST and 3072 for repeated-MNIST, and a balanced test set of size 10,000 for both datasets. All models are trained for up to 30 epochs for MNIST and up to 40 epochs for repeated-MNIST. We sample an initial training set of size 20 (2 per class) and conduct 30 acquisitions of batch size 10 on both datasets, and we use 50 MC dropout samples to estimate the posterior.

The test accuracy and negative log-likelihood (NLL) are shown in Figure 2. ICAL significantly improves the NLL and outperforms all other baselines on accuracy, with higher margins on the earlier acquisition rounds. The performance is consistent across runs (the variance is smaller than other baselines), and is robust even in the repeated-MNIST setup, where all the other greedy methods show worsen performance. We check the frequency that replicas of a single sample were included in acquired batch and as shown in Appendix Figure 8, our method (as well as BatchBALD, BayesCoreset and random) acquired no redundant sample whereas FASS and max entropy acquired up to 3 copies of some samples.

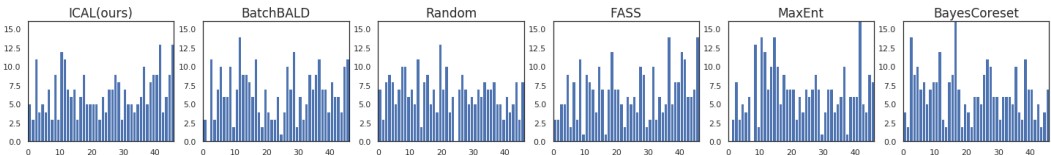

Figure 3: Histogram of the labels of all acquired points using different active learning methods on EMNIST (47 classes). ICAL acquires more diverse and balanced batches while all other methods have overly/under-represented classes.

## 6.2   EMNIST

We then extend the task to a more sophisticated dataset named Extended-MNIST, which consists of 47 classes of 28x28 images of both digits and letters. We used the balanced EMNIST where each class has 2400 training examples. We use a validation set of 16384 and test set of size 18800 (400 per class), and train for up to 40 epochs. We use a CNN consisting of three convolutional layers with 32, 64, and 128 3x3 convolution filters, each followed by MC dropout, 2x2 max-pooling and ReLU. A fully connected layer with 512 hidden units and MC dropout is used after convolutional layers. We use an initial train set of 47 (1 per class) and make 60 acquisitions of batch size 5. 50 MC dropout samples are used to estimate the posterior.

The results are in Figure 4. We do substantially better in terms of both accuracy and NLL compared to all other methods. A clue as to why our method outperforms on EMNIST can be found in Figure 3. ICAL is able to acquire more diversed and balanced batches while all other methods have overly/under-represented classes (note that BatchBALD, Random and MaxEnt each totally miss examples from one of classes). This indicates that our method is much more robust in terms of performance even when the number of classes increases, whereas other alternatives degenerate.

## 6.3   Fashion-MNIST

We also examine ICAL's performance on fashion-MNIST which consists of 10 classes of 28x28 Zalando's article images. We use a validation set of 3072 and test set of size 10000 (1000 per class), and train for up to 40 epochs. The network architecture is the same as the one used in MNIST task. We use an initial train set of 20 (2 per class) and make 30 acquisitions of batch size 10. 100 MC dropout samples are used to estimate the posterior. As shown in Figure 4, we again do significantly better in terms of both accuracy and NLL compared to all other methods. Note that almost all baselines were inferior to random baseline except ICAL, showing the robustness of our method.

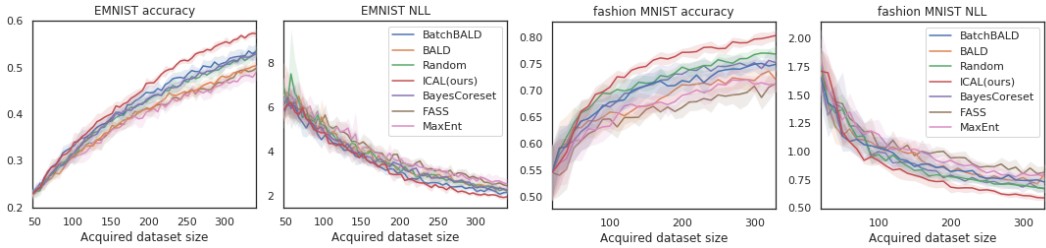

Figure 4: Performance on EMNIST and fashion-MNIST, ICAL significantly improves the accuracy and NLL.

## 6.4 CIFAR

Finally we test our method on the CIFAR-10 and CIFAR-100 datasets Krizhevsky et al. (2009) in a large batch size setting. CIFAR-10 consists of 10 classes with 6000 images per class whereas CIFAR-100 has 100 classes with 600 images per class. We use a validation set of size 1024, and a balanced test set of size 10,000 for both datasets. For CIFAR-10, we start with an initial training set of 10000 examples (1000 per class) while for CIFAR-100, we start with 20000 examples (200 per class). We do 10 acquisitions on CIFAR-10 and 7 acquisitions on CIFAR-100 with batch size of 3000. We use a ResNet18 with additional 2 fully connected layers with MC dropouts, and train for up to 60 epochs with learning rate 0.1 (allow early stopping). We run with 8 different seeds. The results are in Figure 5. Note that we are unable to compare against BatchBALD for either CIFAR dataset as it runs out of memory.

For CIFAR-10, ICAL dominates *all* other methods for all acquisitions except two – when the acquired dataset size is 19000 and when it is 28000. ICAL also has the highest area under curve (auc) for accuracy compared to all other methods; with p-value $\leq 0.007$ except for BALD and Max Entropy for which we have better auc with p-value $0.24, 0.15$ respectively. ICAL also achieves the highest accuracy at the end of all 10 acquisitions. With CIFAR-100, on all acquisitions ICAL outperforms a majority of the methods. Furthermore, ICAL again finishes with the highest accuracy by a significant margin at the end of the acquisition rounds and it again have the highest auc compared to all other methods. Detailed comparison results are in the Appendix Table 2.

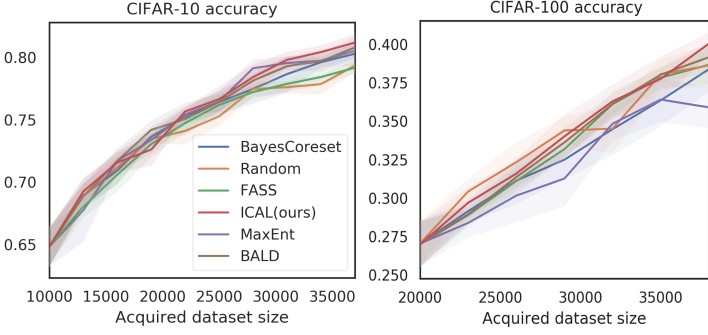

Figure 5: Performance on CIFAR-10 and CIFAR-100 with batch size=3000 using 8 seeds

## 7 Conclusion

We develop a novel batch mode active learning acquisition function ICAL that is model agnostic and applicable to both classification and regression tasks (as it relies on only samples from the posterior predictive distribution). We develop key optimizations that enable us to scale our method to large acquisition batch and unlabeled set sizes. We show that we are robustly able to outperform state of the art methods for batch mode active learning on a variety of image classification tasks in a deep neural network setting.

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

# Appendix

## Motivating example 2

Suppose we have a model distribution with 10 possible models $\omega_1, \ldots, \omega_{10}$ with equal prior probability of being the true model ($p(w_i) = 0.1$ for $\forall i$). Let the datapoints be $x_1, \ldots, x_L$ with their labels taking 4 possible values. We define $p_{ij}^k = p(y_i = j | x_i, \omega_k)$ as the probability of the $j$th class for the $ith$ datapoint given by the $k$th model. Let

$$p_{1j}^k = 1; j = k, 1 \leqslant k \leqslant 3$$
$$p_{14}^k = 1; 4 \leqslant k \leqslant 10$$
$$p_{i1}^k = 1, p_{i2}^{10} = 1; 1 \leqslant k \leqslant 9, 2 \leqslant i \leqslant L$$

| | $\omega_1$ | $\omega_2$ | $\omega_3$ | $\omega_4$ | $\omega_5$ | $\omega_6$ | $\omega_7$ | $\omega_8$ | $\omega_9$ | $\omega_{10}$ |
|---|---|---|---|---|---|---|---|---|---|---|
| $x_1$ | 1 | 2 | 3 | 4 | 4 | 4 | 4 | 4 | 4 | 4 |
| $x_2 \ldots x_L$ | 1 | 1 | 1 | 1 | 1 | 1 | 1 | 1 | 1 | 2 |

Table 1: Labels that the different points $x_i$ take with probability 1 under different models. The columns are the different models $\omega_k$, and the rows are the different points.

Given that we have no other information about the models, we update the posterior probabilities for the models as follows – if a model $\omega_k$ outputs label $l$ for a point $x$ but after acquisition, the label for $x$ is not $l$, then we know that is not the correct model and thus its posterior probability is 0 (so it is eliminated). Otherwise we have no way of distinguishing between the remaining models so they all have equal posterior probability. Then for $x_1$ the mutual information is

$$\mathbb{I}[y_1, \omega | x_1, \mathcal{D}_{train}]$$
$$= H[y_1 | x_1] - \mathbb{E}_{p(\omega | \mathcal{D}_{train})}[H[y_1 | x_1, \omega]] = 0.94$$

For $x_2 \ldots x_L$, $\mathbb{I}[y_{2-L}, \omega | x_{2\ldots L}, \mathcal{D}_{train}] = 0.325$. However selecting $x_1$ would decrease the expected posterior entropy $H[y_{2-L} | x_{2\ldots L}, x_1, y_1, \mathcal{D}_{train}]$ from 0.325 to only 0.287. Acquiring any of $x_{2\ldots L}$ instead of $x_1$, however, would decrease that entropy to 0, which would cause a much larger decrease in the expected posterior entropy averaged over $x_{1\ldots L}$ if $L$ is large enough. The detailed calculations are in the later subsection.

While $x_{2\ldots L}$ may not contribute much to the entropy of the *joint* predictive distribution or to the MI with respect to the model parameters compared to $x_1$, collectively they will be weighted $L - 1$ times more than $x_1$ when looking at the accuracy. We should thus expect a well-calibrated model to have a higher uncertainty, and thus make a lot more errors on $x_{2\ldots L}$, if $x_1$ is acquired versus if any of $x_{2\ldots L}$ are acquired. For instance, in the above example, as $L$ increases, the expected error rate would approach $\approx 0.7 \times (1/7 \times 6/7) \times 2 = 0.17$ (0.7 as 0.3 of the times the value of $x_1$ would also fix what the true model is reducing error rate on all $x$ to 0) if $x_1$ is acquired as the errors for $x_{2\ldots L}$ are correlated, whereas the rate would approach 0 were any of $x_{2\ldots L}$ to be acquired.

## Derivation for Example 2

For $x_1$, the mutual information between the predicted label $y_1$ and model parameters is:

$$\mathbb{I}[y_1, \omega | x_1, \mathcal{D}_{train}]$$
$$= H[y_1 | x_1] - \mathbb{E}_{p(\omega | \mathcal{D}_{train})}[H[y_1 | x_1, \omega]]$$
$$= H[\sum_{k=1}^{10} p(y_1 | x_1, \omega_k) p(\omega_k)] - \sum_{k=1}^{10} p(\omega_k) H[p(y_1 | x_1, \omega_k)]$$
$$= -(3 \times (\frac{1}{10} \times \log(\frac{1}{10})) + \frac{7}{10} \times \log(\frac{7}{10}))$$

$$-10 \times \frac{1}{10} \times (-(1 \times \log(1) + 0 \times \log(0)))$$
$$= 0.940$$

For $x_{2\dots L}$,

$$\mathbb{I}[y_{2-L}, \omega | x_{2\dots L}, \mathcal{D}_{train}]$$
$$= -(\frac{9}{10} \times \log(\frac{9}{10}) + \frac{1}{10} \times \log(\frac{1}{10}))$$
$$- 10 \times \frac{1}{10}(-(1 \times \log(1) + 0 \times \log(0)))$$
$$= 0.325$$

After acquiring $x_1$, assuming the true label for $x_1$ is 1, then we update the posterior over the model parameter such that $p'(w_1)|_{y_1=1} = 1$ and $p'(w_k)|_{y_1=1} = 0$ for $1 < k \leqslant 10$. Then the expected averaged posterior entropy for $x_{1\dots L}$ is:

$$\frac{1}{L-1} \sum_{i=2}^{L} H[y_i|x_i]|_{y_1=1}$$
$$= \frac{1}{L-1} \sum_{i=2}^{L} H[\sum_{k=1}^{10} p(y_i|x_i, \omega_k)p'(\omega_k)|_{y_1=1}]$$
$$= \frac{1}{L-1} \times (L-1) \times (-(1 \times \log(1) + 0 \times \log(0)))$$
$$= 0$$

Similarly, we could compute the case where the true label for $x_1$ is 2-4:

$$\frac{1}{L-1} \sum_{i=2}^{L} H[y_i|x_i]|_{y_1=2} = 0$$

$$\frac{1}{L-1} \sum_{i=2}^{L} H[y_i|x_i]|_{y_1=3} = 0$$

$$\frac{1}{L-1} \sum_{i=2}^{L} H[y_i|x_i]|_{y_1=4}$$
$$= \frac{1}{L-1} \times (L-1) \times (-(\frac{6}{7} \log(\frac{6}{7}) + \frac{1}{7} \log(\frac{1}{7})))$$
$$= 0.41$$

The expectation of the averaged posterior entropy with respect to predicted label for $y_1$ (since we don't know the true label) is:

$$H[y_{2-L}, \omega | x_{2\dots L}, x_1, y_1 \mathcal{D}_{train}]$$
$$= \mathbb{E}_{y_1 \sim p(y_1|\mathcal{D}_{train})}[\frac{1}{L-1} \sum_{i=2}^{L} H[y_i|x_i]|_{y_1}]$$
$$= \frac{1}{10} \times 0 + \frac{1}{10} \times 0 + \frac{1}{10} \times 0 + \frac{7}{10} \times 0.41$$
$$= 0.287$$

## Baseline acquisition function details

**Max entropy** selects the points that maximize the predictive entropy

$$\alpha(x, \mathcal{M}) = H(y|x, \mathcal{D}_{train})$$

$$= -\sum_{c} p(y = c|x, \mathcal{D}_{train}) \log(p(y = c|x, \mathcal{D}_{train}))$$

**BatchBALD**    BatchBALD (Kirsch et al., 2019) tries to find a batch of points that has the highest mutual information with respect to the model parameters. **BALD** is the non-batched version of BatchBALD. Formally

$$\alpha_{BatchBALD}(\{x_1, \ldots, x_B\}, p(\omega))$$
$$= H(y_1, \ldots, y_B) - \mathbb{E}_{p(\omega)}[H(y_1, \ldots, y_B | \omega)]$$

**Filtered active submodular selection (FASS)**    FASS (Wei et al., 2015) samples the $\beta \times B$ most uncertain points $\mathcal{B}'$ and then subselect $B$ points that are as representative of $\mathcal{B}'$ as possible. For the measure of uncertainty, FASS uses entropy $H(y|x, \mathcal{D}_{train})$. To measure the representativeness of $\mathcal{B}$ to $\mathcal{B}'$, FASS tries to choose $\mathcal{B}$ to maximize the following function

$$f(\mathcal{B}) = \sum_{y \in \mathcal{Y}} \sum_{i \in V^y} \max_{s \in \mathcal{B} \cap V^y} w(i, s)$$

Here $V^y \subseteq \mathcal{B}'$ is the set of points in $\mathcal{B}'$ with predicted label, $y$ and $w(i, s) = d - ||x_i - x_s||_2^2$ is the similarity function between points indexed by $i, s$ where $x_i, x_s \in \mathcal{X}$ and $d$ is the maximum distance between two points. The idea here is that if a point in $\mathcal{B}$ already exists that is close to some point $x' \in \mathcal{B}'$, then $f(\mathcal{B})$ will favor adding points to the batch that are close to points other than $x'$, thus increasing the batch diversity. Note that FASS is equivalent to Max Entropy if $\beta = 1$.

**Bayesian Coresets**    In Pinsler et al. (2019), they try to build a batch such that the log posterior after acquiring that batch best approximates the complete data log posterior (i.e. the log posterior after acquiring the entire pool set). Their approach closely follows the general Bayesian Coreset (Campbell and Broderick, 2018) approach which constructs a weighted subset of data that approximates the full dataset. Crucially (Pinsler et al., 2019) assume that the posterior predictive distribution $Y_p$ of a point $p$ is independent of that of the corresponding distribution $Y_{p'}$ of another point $p'$ – an assumption we do not make. We show in the next section why avoiding such an assumption lets us more effectively minimize the error with respect to the test distribution versus just optimizing for maxmizing information gain for the model posterior. As (Pinsler et al., 2019) require a variable batch size whereas all other methods (including ours) use a fixed batch size, for fairness of comparison, if the batch for this approach is smaller than the batch size being used, we fill the rest of the batch with random points. In practice, we only observe this being necessary for CIFAR.

**Random**    The points are selected uniformly at random from the unlabeled pool. Thus $\alpha(x, \mathcal{M})$ is the uniform distribution.

## Further statistical background

A divergence $\Lambda$ between two distributions is a measure of the discrepancy or difference between two distributions $P, Q$. A key property of a divergence is that it is 0 if and only if $P, Q$ are the same distribution. In this paper, we will be using the KL divergence and the MMD, which are respectively defined as

$$D_{KL}(P||Q) = - \sum_{x \in \mathcal{X}} P(x) \log(\frac{Q(x)}{P(x)})$$
$$MMD_k^2(P, Q) = \mathbb{E}k(X, X') + k(Y, Y') - 2k(X, Y)$$

where $k$ is a kernel in the Reproducing Kernel Hilbert Space (RKHS) $\mathcal{H}$ and $\mu_k$ is the mean embedding of the distribution into $\mathcal{H}$ as per the kernel $k$. We can then use the notion of divergence to define the *dependency* $\mathfrak{d}$ between a set of random variables $X_{1:n}$ as follows

$$\mathfrak{d}(X_{1:n}) = \Lambda(P_{1:n}, \otimes_i P_i)$$

where $P_{1:n}$ is the joint distribution of $X_{1:n}$, $P_i$ the marginal of $X_i$ with $\otimes P_i$ being the product of marginals. For $D_{KL}$ the dependency is exactly MI as defined above. For $MMD$ the dependency is the Hilbert-Schmidt Independence Criterion ($HSIC$).

## Hilbert-Schmidt Independence Criterion (HSIC)

Formally, if $X, Y$ are drawn from the joint distribution $P_{XY}$, then their $HSIC$ is defined as –

$$
\begin{aligned}
HSIC(P_{XY}, k, l) = \mathbb{E}_{x,x',y,y'}[k(x,x')l(y,y')] \\
+ \mathbb{E}_{x,x'}[k(x,x')]\mathbb{E}_{y,y'}[l(y,y')] \\
- 2\mathbb{E}_{x,y}[\mathbb{E}_{x'}[k(x,x')]\mathbb{E}_{y'}[k(y,y')]]
\end{aligned}
$$

where $(x,y)$ and $(x',y')$ are independent pairs drawn from $P_{XY}$. Note that $HSIC(P_{XY}) = 0$ if and only if $P_{XY} = P_X P_Y$, that is, if $X, Y$ are independent, for chracteristic kernels $k$ and $l$.

For the case where we are measuring the *joint* dependence between $d$ variables, we can use the $HSIC$ statistic (Sejdinovic et al., 2013a; Pfister et al., 2018). The computational complexity of $HSIC$ is bounded by the time taken to compute the kernel matrix which is $O(m^2 d)$ where $m$ is the number of samples and $d$ the number of random variables. We use $\widehat{HSIC}$ to denote the empirical estimator of the $HSIC$ statistic.

## Proof of Proposition 1

$k^*$ is positive semidefinite (psd) and symmetric as the sum of psd symmetric matrices is also psd symmetric.

## Proof of Proposition 2

We show here that

$$
d\widehat{HSIC}(k^1, k^3, \dots, k^d) + d\widehat{HSIC}(k^2, k^3, \dots, k^d)
$$
$$
= d\widehat{HSIC}(k^1 + k^2, k^3, \dots, k^d)
$$

but the extension to the arbitrary sums is straightforward. Here $d\widehat{HSIC}$ is the estimator for $dHSIC$ which is the $d$-variable version of $HSIC$. It is defined as

$$
dHSIC = \frac{1}{n^2}\sum_{a=1}^{n}\sum_{b=1}^{n}\Pi_{j=1}^{d}k^j(X_{i_a}^j, X_{i_b}^j) + \frac{1}{n^{2d}}\Pi_{j=1}^{d}\sum_{a=1}^{n}\sum_{b=1}^{n}k^j(X_{i_a}^j, X_{i_b}^j) - \frac{2}{n^{d+1}}\sum_{a=1}^{n}\Pi_{j=1}^{d}\sum_{b=1}^{n}k^j(X_{i_a}^j, X_{i_b}^j)
$$

where $k^j$ is the kernel of the $j$th random variable and $X_i^j$ is the $i$th observation for the $j$th random variable. The estimator $d\widehat{HSIC}$ is defined as (Sejdinovic et al., 2013a)

$$
d\widehat{HSIC} = \frac{1}{n^2}\sum_{a=1}^{n}\sum_{b=1}^{n}\Pi_{j=1}^{d}k^j(x_{i_a}^j, x_{i_b}^j) + \frac{1}{n^{2d}}\Pi_{j=1}^{d}\sum_{a=1}^{n}\sum_{b=1}^{n}k^j(x_{i_a}^j, x_{i_b}^j) - \frac{2}{n^{d+1}}\sum_{a=1}^{n}\Pi_{j=1}^{d}\sum_{b=1}^{n}k^j(x_{i_a}^j, x_{i_b}^j)
$$

As $dHSIC$ reduces to $HSIC$ when $d = 2$, the proof for $HSIC$ also follows. Using the definition of $d\widehat{HSIC}$ above,

$$\widehat{dHSIC}(k^1, k^3, \ldots, k^d) + \widehat{dHSIC}(k^2, k^3, \ldots, k^d) =$$

$$\frac{1}{n^2} \sum_{a=1}^{n} \sum_{b=1}^{n} k^1(x_{i_a}^1, x_{i_b}^1) \prod_{j=3}^{d} k^j(x_{i_a}^j, x_{i_b}^j)$$

$$+ \frac{1}{n^{2d}} \sum_{a=1}^{n} (\sum_{b=1}^{n} k^1(x_{i_a}^1, x_{i_b}^1)) \prod_{j=3}^{d} \sum_{b=1}^{n} k^j(x_{i_a}^j, x_{i_b}^j)$$

$$- \frac{2}{n^{d+1}} (\sum_{a=1}^{n} \sum_{b=1}^{n} k^1(x_{i_a}^1, x_{i_b}^1)) \prod_{j=3}^{d} \sum_{a=1}^{n} \sum_{b=1}^{n} k^j(x_{i_a}^j, x_{i_b}^j)$$

$$+ \frac{1}{n^2} \sum_{a=1}^{n} \sum_{b=1}^{n} k^2(x_{i_a}^2, x_{i_b}^2) \prod_{j=3}^{d} k^j(x_{i_a}^j, x_{i_b}^j)$$

$$+ \frac{1}{n^{2d}} \sum_{a=1}^{n} (\sum_{b=1}^{n} k^2(x_{i_a}^2, x_{i_b}^2)) \prod_{j=3}^{d} \sum_{b=1}^{n} k^j(x_{i_a}^j, x_{i_b}^j)$$

$$- \frac{2}{n^{d+1}} (\sum_{a=1}^{n} \sum_{b=1}^{n} k^2(x_{i_a}^2, x_{i_b}^2)) \prod_{j=3}^{d} \sum_{a=1}^{n} \sum_{b=1}^{n} k^j(x_{i_a}^j, x_{i_b}^j)$$

$$= \Big[ \frac{1}{n^2} \sum_{a=1}^{n} \sum_{b=1}^{n} k^1(x_{i_a}^1, x_{i_b}^1) \prod_{j=3}^{d} k^j(x_{i_a}^j, x_{i_b}^j)$$

$$+ \frac{1}{n^2} \sum_{a=1}^{n} \sum_{b=1}^{n} k^2(x_{i_a}^2, x_{i_b}^2) \prod_{j=3}^{d} k^j(x_{i_a}^j, x_{i_b}^j) \Big]$$

$$+ \Big[ \frac{1}{n^{2d}} \sum_{a=1}^{n} (\sum_{b=1}^{n} k^1(x_{i_a}^1, x_{i_b}^1)) \prod_{j=3}^{d} \sum_{b=1}^{n} k^j(x_{i_a}^j, x_{i_b}^j)$$

$$+ \frac{1}{n^{2d}} \sum_{a=1}^{n} (\sum_{b=1}^{n} k^2(x_{i_a}^2, x_{i_b}^2)) \prod_{j=3}^{d} \sum_{b=1}^{n} k^j(x_{i_a}^j, x_{i_b}^j) \Big]$$

$$- \Big[ \frac{2}{n^{d+1}} (\sum_{a=1}^{n} \sum_{b=1}^{n} k^1(x_{i_a}^1, x_{i_b}^1)) \prod_{j=3}^{d} \sum_{a=1}^{n} \sum_{b=1}^{n} k^j(x_{i_a}^j, x_{i_b}^j)$$

$$+ \frac{2}{n^{d+1}} (\sum_{a=1}^{n} \sum_{b=1}^{n} k^2(x_{i_a}^2, x_{i_b}^2)) \prod_{j=3}^{d} \sum_{a=1}^{n} \sum_{b=1}^{n} k^j(x_{i_a}^j, x_{i_b}^j) \Big]$$

$$= \frac{1}{n^2} \sum_{a=1}^{n} \sum_{b=1}^{n} (k^1(x_{i_a}^1, x_{i_b}^1) + k^2(x_{i_a}^2, x_{i_b}^2)) \prod_{j=3}^{d} k^j(x_{i_a}^j, x_{i_b}^j)$$

$$+ \frac{1}{n^{2d}} \sum_{a=1}^{n} \Big[ \sum_{b=1}^{n} (k^1(x_{i_a}^1, x_{i_b}^1)$$

$$+ k^2(x_{i_a}^2, x_{i_b}^2)) \Big] \prod_{j=3}^{d} \sum_{b=1}^{n} k^j(x_{i_a}^j, x_{i_b}^j)$$

$$- \frac{2}{n^{d+1}} \Big[ \sum_{a=1}^{n} \sum_{b=1}^{n} (k^1(x_{i_a}^1, x_{i_b}^1)$$

$$+ k^2(x_{i_a}^2, x_{i_b}^2)) \Big] \prod_{j=3}^{d} \sum_{a=1}^{n} \sum_{b=1}^{n} k^j(x_{i_a}^j, x_{i_b}^j)$$

$$= \widehat{dHSIC}(k^1 + k^2, k^3, \ldots, k^d)$$

## 7.1 Further scaling to large batch sizes

To scale to large batch sizes, instead of adding points to the batch to be acquired one at a time, we can add points in minibatches of size $L$. While this comes at the cost of possible diversity in the batch, we find that the tradeoff is acceptable for the datasets we experimented with. This gives a final computation cost of $O(\frac{|\mathcal{D}_U| m^2 B \cdot C}{L})$ where $C$ is the number of classes. By contrast the corresponding runtime for BatchBALD is $O(\mathcal{D}_U) \cdot B \cdot C \cdot m \cdot m'$ where $m'$ is the number of sampled configurations of $y_{1:n-1}$. For all experiments with ICAL, we were able to use $L = 1$ without any scaling difficulties. For ICAL-pointwise, we used $L = \frac{B}{15}$ only for CIFAR-10 and CIFAR-100. As alluded to previously, ICAL-pointwise can accommodate much larger $L$ compared to ICAL before its performance degrades, allowing for much greater scaling. We evaluate this aspect of ICAL-pointwise in the Appendix.

The final algorithm is given in Algorithm 1.

## 7.2 Algorithm

---
**Algorithm 1** Information Condensing Active Learning (ICAL) $(\mathcal{M}, T, \mathcal{D}_{train}, \mathcal{D}_U, B, K, r, L)$
---

Train $\mathcal{M}$ on $\mathcal{D}_{train}$
**repeat**
    $\mathcal{B} = \{\}$
    **while** $|\mathcal{B}| < B$ **do**
        $Y^U$ = the predictive distribution for $x \in \mathcal{D}_U$ according to $\mathcal{M}$
        $R$ = Set of $r$ randomly selected points from $\mathcal{D}_U$
        $x' = \text{argmax}_x\, \alpha_{ICAL}(\mathcal{B} \cup \{x\}, HSIC)$ with the optimizations as specified in Section 5.1
        and 5.2
        $\mathcal{B} = \mathcal{B} \cup \{x'\}$
    **end while**
    $\mathcal{D}_{train} = \mathcal{D}_{train} \cup \mathcal{B}$
    Retrain $\mathcal{M}$ on $\mathcal{D}_{train}$
**until** $T$ iterations reached
Return $\mathcal{M}$

---

## ICAL-pointwise

To evaluate the marginal dependency increase if a candidate point $x$ is added to batch $\mathcal{B}$, we sample a set $R$ from the pool set $\mathcal{D}_U$ and compute the pairwise $dHSIC$ of both $\mathcal{B}$ and $\mathcal{B}' = \mathcal{B} \cup \{x\}$ with respect to each point in $R$. Let the resulting vectors (each of length $|R|$) with the $dHSIC$ scores be $\mathfrak{d}_\mathcal{B}$ and $\mathfrak{d}_{\mathcal{B}'}$. Then the marginal dependency increase statistic $M_x$ for point $p$ is $M_x = \frac{1}{|R|} \sum_i \max((\mathfrak{d}_{\mathcal{B}'}^i/\mathfrak{d}_\mathcal{B}^i), 1)$ where $i$ is the $ith$ element of the vector. When then modify the $\alpha_{ICAL}$ as follows - $\alpha'_{ICAL}(\mathcal{B} \cup \{x\}) = \alpha_{ICAL}(\mathcal{B} \cup \{x\}) \cdot (M_x - 1)$ and use the point with the highest value of $\alpha'_{ICAL}$ as the point to acquire. Note that as we want to get as accurate an estimate of $M_x$ as possible, we ideally want to choose as large a set $R$ as possible. In general, we also want to choose $|R|$ to be greater than the number of classes. This makes ICAL-pointwise more memory intensive compared to ICAL. We also tried another criterion for batch selection based on the minimal-redundancy-maximal-relevance Peng et al. (2005) but that had significantly worse performance compared to ICAL and ICAL-pointwise.

In Figure 6, we analyze the performance of ICAL versus ICAL-pointwise when their parameters are set such that computational cost is about the same. As can be seen they are broadly similar with ICAL-pointwise having a slight advantage in earlier acquisitions and ICAL being slightly better in later ones.

We also analyze the relative performance as the mini-batch size $L$ changes in Figure 7. In the Figure, $iter = \frac{B}{L}$ is the number of iterations taken to build the entire acquisition batch (note that the actual acquisition happens *after* the entire batch has been built). ICAL-pointwise requires more computation

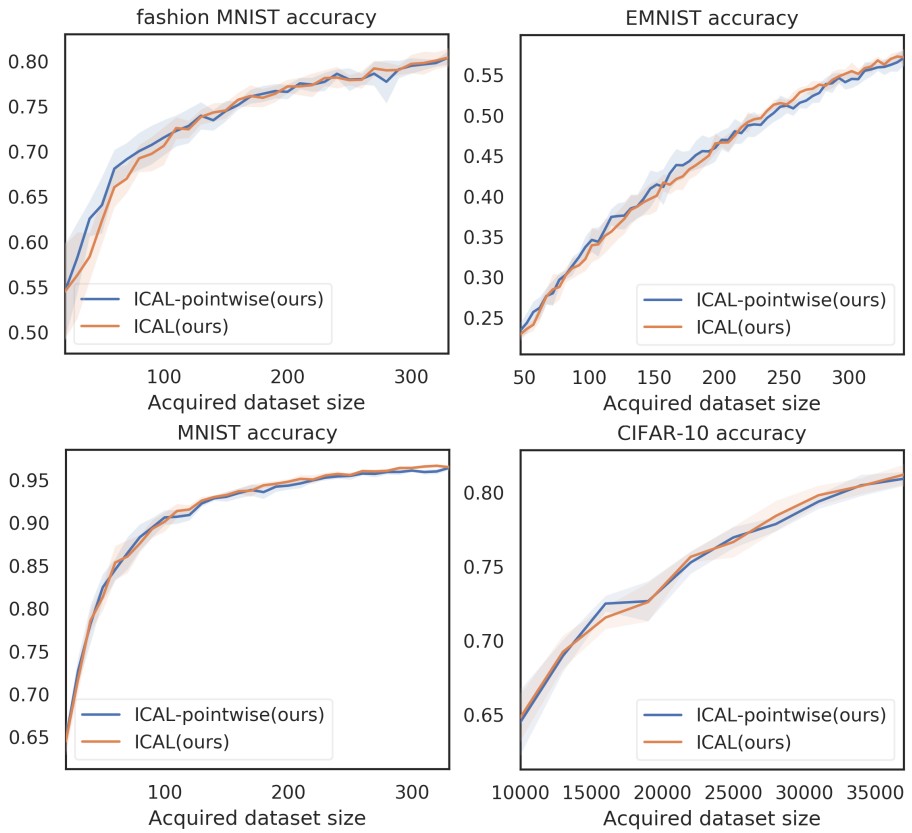

Figure 6: Relative performance of ICAL and ICAL-pointwise on smaller datasets (EM-NIST,FashionMNIST,MNIST and CIFAR10) with parameters set to equivalent computation cost

time than ICAL in small $L$ setup, however if time is the major constraint, ICAL-pointwise is to be preferred as its performance degrades more slowly as $L$, the size of the minibatch, increases. As the performance usually peaks at $L = 1$, if one is trying to get the best performance or if memory is a constraint, then ICAL is to be preferred.

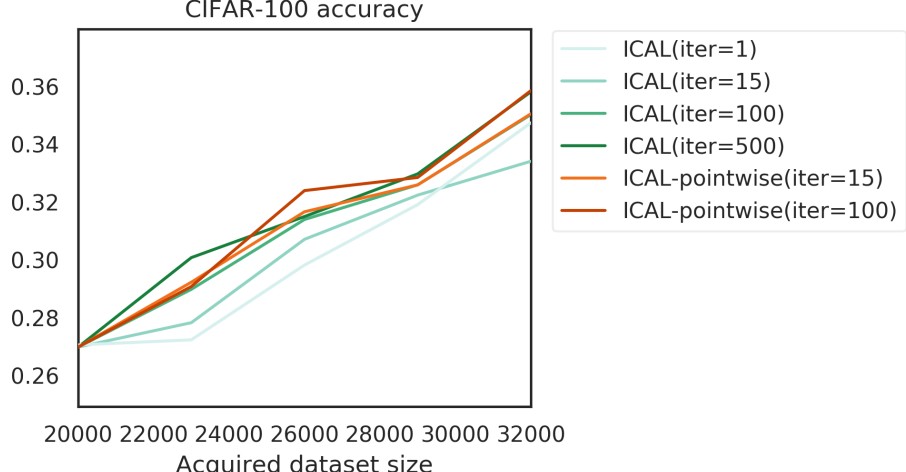

Figure 7: Relative performance of ICAL and ICAL-pointwise on CIFAR100 with different mini-batch size $L$. $iter = \frac{B}{L}$ is the number of iterations taken to build the entire acquisition batch of size $B$ (note that the actual acquisition happens *after* the entire batch has been built)

### Diversity of acquired samples in repeated-MNIST

To check if ICAL's acquisition batches are diversed enough, we plot the number of times different number of copies of a same sample has been acquired by each method. As shown in figure 8, our method (as well as BatchBALD, BayesCoreset and Random) successfully avoided acquiring redundant copies of the same sample, whereas FASS and Max Entropy acquired up to 3 copies of the same replica in most acquisitions. This proves that the batched active learning strategies are better in diversity.

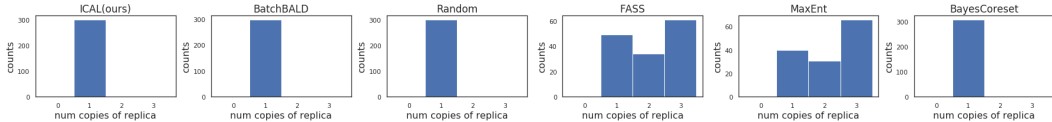

Figure 8: Frequencies where different numbers of copies (1-3) of a same sample has been acquired by each method.

### Further CIFAR-10 and CIFAR-100 results

| | CIFAR-10 | | CIFAR-100 | |
|---|---|---|---|---|
| | ICAL outperformance $p$-value | AUC of accuracy curve | ICAL outperformance $p$-value | AUC of accuracy curve |
| Random | $6.0e-05$ | $0.742 \pm 0.004$ | $0.444$ | $\mathbf{0.338 \pm 0.007}$ |
| BALD | $0.248$ | $0.751 \pm 0.004$ | $0.124$ | $0.335 \pm 0.003$ |
| FASS | $8.7e-06$ | $0.742 \pm 0.003$ | $0.053$ | $0.333 \pm 0.005$ |
| BayesCoreset | $0.007$ | $0.748 \pm 0.003$ | $0.003$ | $0.327 \pm 0.007$ |
| MaxEnt | $0.158$ | $0.751 \pm 0.003$ | $8.5e-05$ | $0.321 \pm 0.007$ |
| ICAL (ours) | N/A | $\mathbf{0.753 \pm 0.003}$ | N/A | $\mathbf{0.338 \pm 0.005}$ |

Table 2: Area under curve of the accuracy curve for the different methods on CIFAR-10/CIFAR-100 and p-values of the ICAL out-performance significance when compared to each of the methods. Highest AUC values of each task are highlighted in **bold**.

Further CIFAR results are in Table 2. For CIFAR-100, Random has a high p-value but that is mainly because it performs a bit better in the beginning vs. all other methods but its performance quickly degrades and it is far below ICAL in the final iteration.

### Runtime and memory considerations

BatchBALD runs out of memory on CIFAR-10 and CIFAR-100 and thus we are unable to compare against it for those two datasets. For the MNIST-variant datasets, ICAL takes about a minute for building the batch to acquire (batch sizes of 5 and 10). For CIFAR-10 (batch size 3000), with $L = 1$, the runtime is about 20 minutes but it scales linearly with $1/L$ (Figure 10). Thus it is only 5 minutes for $L = 30$ ( $iter = 100$) which is already sufficient to give comparable performance to $L = 1$ (Figure 9). For CIFAR-100 (batch size 3000), the performance does degrade with high $L$ but as we mentioned previously, ICAL-pointwise holds up a lot better in terms of performance with high $L$ (Figure 7) and thus if time is a strong consideration, that variant should be used instead.

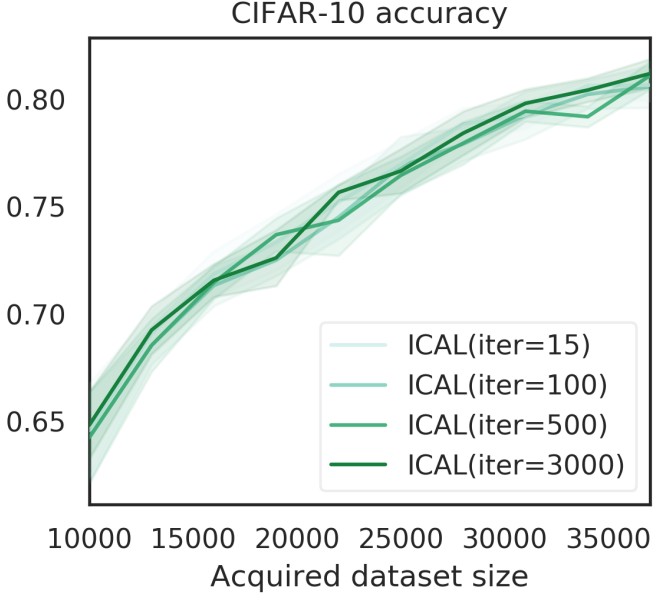

Figure 9: CIFAR10 performance with different L. $iter = \frac{B}{L}$ is the number of iterations taken to build the entire acquisition batch of size $B$ (note that the actual acquisition happens *after* the entire batch has been built)

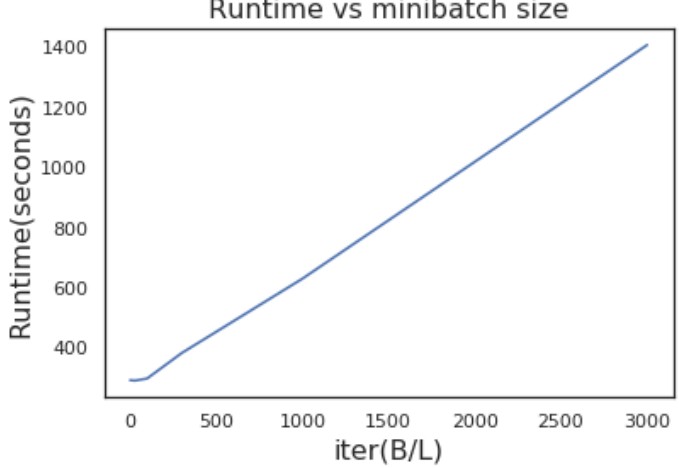

Figure 10: Runtime of ICAL on CIFAR10 with different minibatch size $L$.