# OpenReview forum: "Information Condensing Active Learning"
_ICLR.cc/2021/Conference — Reject_

### Official Review · AnonReviewer1 · 2020-10-20
**An interesting, principled new active learning method**

**Rating:** 6
**Confidence:** 3

**Review:**

The authors propose a novel acquisition function for active learning based on the (sample) Hilbert Schmidt Independence Criterion (HSIC) statistic. The authors use several standard approximations (using MC dropout to sample from the posterior, building the pool batches greedily, subsampling other unlabelled points when estimating the value of a candidate batch) as well as useful properties of the HSIC statistic, to make the proposed method computationally tractable.

Pros:

The proposed method is fairly straightforward and uses relatively well studied tools.

Section 4 provides a valuable analysis of existing active learning methods which is of independent interest, and shows how the new method is not as susceptible to the flaws of other methods. This insight can be quite valuable to further develop new active learning algorithms.

Figure 3 suggests a reason for why the proposed method outperforms existing ones on some data sets (although it would be interesting to get statistics which back up the claim that individual batches are more balanced as well as the overall sampling process).

Experiments compare the new method to a reasonable set of SOTA active learning methods.

Cons:

The new method is competitive on all experiments, but can only be claimed to outperform existing methods on EMNIST and fashion-MNIST.

Would be interesting to see things like robustness to some of the approximation choices (number of MC dropout resamples for example).

The ideas and observations here are a contribution to the active learning community, and experiments still show this method to be competitive with existing methods. Therefore I currently lean towards accepting.

---

> ### Author Response · Authors · 2020-11-23
> **Re:AnonReviewer1**
>
> We thank the reviewer for considering our method novel and computationally tractable and the experiments reasonable, and finding our work of independent interest to the community.
>
> **The new method is competitive on all experiments, but can only be claimed to outperform existing methods on EMNIST and fashion-MNIST.**
>
> Although it might be difficult to judge visually from the plots, we do outperform existing methods on CIFAR-10/100 with statistically significance against the majority of the competing methods. We show the statistics in Table 2 in the Appendix.
>
> **Would be interesting to see things like robustness to some of the approximation choices (number of MC dropout resamples for example).**
>
> We thank the reviewer for the suggestion. We ran some preliminary experiments which indicated that there was little difference in accuracy beyond 50 resamples for the datasets we looked at in the paper. We want to note that a decrease in the number of resamples would also affect other methods and so we don’t expect the relative ordering of the performance of the methods to change much.

---

### Official Review · AnonReviewer2 · 2020-10-27
**Another acquisition function that seems to do a little better on several datasets for CNNs**

**Rating:** 6
**Confidence:** 4

**Review:**

This paper introduces a model agnostic active learning technique that maximizes the dependency between a batch and the rest of the unlabeled pool. The dependence is measured via the Hilbert Schmidt Independence Criterion and this paper introduces some approximations/optimizations to speed up the method.

In terms of novelty, this paper introduces another acquisition function, which to the best of my knowledge is technically novel. However, the reasoning or intuition for the algorithm does not seem very novel. For instance, expected error reduction (EER) is motivated by a similar intuition.

The significance of this paper rests on its empirical evaluation as there isn't any justification for the heuristic algorithm beyond a conceptual motivation (which isn't unusual). The empirical results are relatively strong, comparing to a variety of recent algorithms an outperforming the others by a small margin (and a larger margin on a couple of datasets).

It would have been nice to see the model agnosticity used by running experiments with different models beyond CNNs. In fact, if some other methods (which apply to CNNs) are discounted because they are not model agnostic, the experiments should really include these other methods or include more than just CNNs. It would also have been nice to see the learning curves in the higher accuracy range, which may be more practical: for instance, in addition to the current plots, also having plots with larger batches that go up to budgets of 1000 or more.

AFTER READING AUTHOR RESPONSE-----------------------------------------------------------------------------------------

Thanks for the response.

Ideally, you would be able to show results for ensembles of decision trees or some model that is very different from neural networks.

---

> ### Author Response · Authors · 2020-11-23
> **Re:AnonReviewer2**
>
> We appreciate the reviewer’s comments and we thank the reviewer for recognizing our acquisition function novel and finding our empirical evaluation significant and strong. We would like to respond to some of the reviewers concerns below:
>
> **It would have been nice to see the model agnosticity used by running experiments with different models beyond CNNs. In fact, if some other methods (which apply to CNNs) are discounted because they are not model agnostic, the experiments should really include these other methods or include more than just CNNs.**
>
> We agree that experiments with more models would be interesting and thank the reviewer for the suggestion! We used CNN models because they were the most suitable architecture for the benchmarking image datasets, which were also used by the other baselines. We did try using diverse architectures for different experiments (e.g. varying number of convolutional layers, and using resnet architecture for CIFAR). Unfortunately due to time and computational capacity constraints we are unable to include additional experiments with completely different types of models/datasets, we will try to add in such results for the camera ready.
>
> **It would also have been nice to see the learning curves in the higher accuracy range, which may be more practical: for instance, in addition to the current plots, also having plots with larger batches that go up to budgets of 1000 or more.**
>
> We do have results for large batches -- in particular, CIFAR-10/100 use a batch size of 3000, and we ran acquisitions for reasonably long such that the performance is close to the best a model can achieve with the full dataset used as training set. For the MNIST datasets, we ran 30 acquisitions and observed the model performance almost converged (accuracy >0.95) at the end of the acquisitions.

---

### Official Review · AnonReviewer3 · 2020-10-28
**A novel batch acquisition function with promising results---reservations arise about its exposition in the paper, however**

**Rating:** 6
**Confidence:** 3

**Review:**

The paper introduces a new acquisition function for (batch) active learning that uses a variant of the Hilbert Schmidt Independence Criterion to acquire diverse batches that are informative for the remainder of the unlabeled set. The new acquisition function leads to better calibration, as evidenced by achieving lower NLL (for similar accuracy), as well as better accuracy on EMNIST and FashionMNIST than current baselines.

Overall, I’m scoring the paper with a weak reject. While the paper provides good results on EMNIST and FashionMNIST, it does not seem to scale to CIFAR-10 that well. More importantly, the reviewer had trouble understanding the description of the approach in the paper. The novel idea of using HSIC instead of estimating mutual information terms, which scale badly in batch size, is very interesting and needs to be explored. Sadly, for this reviewer at least, the exposition in the paper is very confusing and hard to follow to the point that this reviewer cannot recommend acceptance in its current form.

### Strengths

The motivation of the paper is strong: it points out that current methods neglect to look at the effect of sample selection on the unlabelled set and focus instead on informativeness for the current model. This prevents current acquisition functions from taking the distribution of the unlabelled set into account. Section 4 provides two convincing examples for this. The simpler one being a strong class imbalance.

The paper proposes to estimate a d-variable Hilbert Schmidt Independence Criterion (dHSIC from Pfister 2018) instead of a mutual information term that includes a d-variable joint distribution (like in BatchBALD by Kirsch 2019). Computation of joint entropies does not scale well, and thus using dHSIC seems like a good alternative to capture dependencies and which also scales better.

The paper provides strong empirical evidence that its approach works much better on EMNIST and FashionMNIST. It provides performance comparable or slightly better to BatchBALD on MNIST and RepeatedMNIST, and it also shows that overall, it seems to pick from different classes more uniformly.

### Concerns

1. This reviewer has had difficulties understanding the method described in the paper as well as its practical implementation. HSIC and dHSIC are not properly introduced in the main paper, whereas mutual information, which is not used, is defined. dHSIC is not even explicitly defined in the appendix. Moreover, the step from having a (MC-Dropout-based) BNN with sampled output probabilities for each pool sample to computing kernels for different variables is not clear. Is consistent MC dropout used? How are the kernels computed?
2. The paper mentions that the pool set is subsampled for computing the pool kernel matrix (acc to Proposition 1) and this noise helps to acquire diverse batches. From the reviewer’s experience, methods that benefit from noise usually perform worse for imbalanced datasets. Could an ablation be run to show the trade-off? Especially given the examples in the motivation in Section 4, another experiment with imbalanced classes would be helpful to show that the proposed approach helps with the issues detailed in its motivation.
3. This reviewer is also confused about the runtime complexity of the proposed algorithm. In the appendix, $dHSIC(x_1, \ldots, x_d)$ has runtime complexity $O(n^2 d)$, where $n$ is the number of samples (the reviewer is guessing this) and $d$ is the number of variables. The number of classes seems to be missing. It has to be linear in the classes and number of MC samples at least as the model outputs need to be used at some point. BatchBALD’s runtime complexity is also stated incorrectly, see Kirsch 2019.
4. In general, the paper is not clearly written, with a notation that changes between the main paper and appendix and variables whose meaning is not explained in the text: for example, the proof of Proposition 2, or the definition of HSIC in the appendix, which does not explain from which distributions the variables are sampled specifically.

This reviewer was not well-acquainted with kernel methods and RHKS.

---
UPDATE: After reading the replies, I have updated my score from 5 to 6.

---

> ### Author Response · Authors · 2020-11-23
> **Re:AnonReviewer3**
>
> We thank the reviewer for finding the idea of HSIC novel and interesting, and for recognizing our positive results on EMNIST and FashionMNIST. We ran some additional analysis to address part of the concerns that the reviewer raised and we would like to respond as following:
>
> **While the paper provides good results on EMNIST and FashionMNIST, it does not seem to scale to CIFAR-10 that well**
>
> While our results on EMNIST and FashionMNIST are stronger, we would like to highlight that ICAL also outperforms all other methods on CIFAR-10 as well as CIFAR-100 with the outperformance being statistically significant for the majority of the methods. As it might be hard to tell from the noisy plots visually,  we included Appendix Table 2 with the Area under curve and statistical significance of ICAL’s outperformance on CIFAR datasets.
>
> **This reviewer has had difficulties understanding the method described in the paper as well as its practical implementation. HSIC and dHSIC are not properly introduced in the main paper, whereas mutual information, which is not used, is defined. dHSIC is not even explicitly defined in the appendix.**
>
> We define HSIC in Section 3 and our algorithm uses only HSIC. We do mention dHSIC in the appendix (and cite the appropriate papers) to make some proofs easier. We apologize for not having a proper definition of dHSIC in the appendix itself and have added it in the updated version.
>
> **Moreover, the step from having a (MC-Dropout-based) BNN with sampled output probabilities for each pool sample to computing kernels for different variables is not clear. Is consistent MC dropout used? How are the kernels computed?**
>
> If by consistent MC dropout, the reviewer means whether we use the same set of dropout masks for the entire pool set, then yes. In particular we do the following -- let’s say we have $C$ classes and we take m MC-dropout samples for $N$ examples. This gives us a tensor of probabilities for each class for each MC-dropout sample for each point. The dimensions of the tensor are $N\times m\times C$. For each classification probability vector, we take a sample from that categorical probability distribution to get a new tensor also of dimensions $N\times m\times C$ (note that we don’t have to do that and can just work with the probability tensor but the results are slightly better if we take a sample). For each example point, we now have a matrix of dimensions $mC$. We then compute the pairwise euclidean distance between the m vectors (each of dimension $C$) to get a new distance matrix of size $m \times m$. We then pass this distance matrix through the kernel function to get a kernel matrix of size again $m\times m$.
>
> For a batch kernel matrix, we average together the kernel matrices of the individual examples constituting the batch.
>
> **The paper mentions that the pool set is subsampled for computing the pool kernel matrix (acc to Proposition 1) and this noise helps to acquire diverse batches. From the reviewer’s experience, methods that benefit from noise usually perform worse for imbalanced datasets. Could an ablation be run to show the trade-off? Especially given the examples in the motivation in Section 4, another experiment with imbalanced classes would be helpful to show that the proposed approach helps with the issues detailed in its motivation.**
>
> We ran some preliminary experiments on CIFAR-10 with imbalanced unlabeled set (with 4 of the classes downsampled to 1/3rd size). Due to computational constraints, we were only able to run ICAL, BayesCoreset, MaxEnt and BALD for 6 iterations with 6 random seeds and find that again we do better on most iterations. In terms of Area Under Curve (AUC), we (AUC=0.751) outperform BayesCoreset (AUC=0.743) and MaxEnt (AUC=0.740) and are competitive with BALD (AUC=0.752). As mentioned in the paper, we are unable to compare against BatchBALD for CIFAR datasets as it runs out of memory
>
> **This reviewer is also confused about the runtime complexity of the proposed algorithm. In the appendix, $dHSIC(x_1,\dots,x_d)$ has runtime complexity $O(n^2 d)$, where $n$ is the number of samples (the reviewer is guessing this) and $d$ is the number of variables. The number of classes seems to be missing. It has to be linear in the classes and number of MC samples at least as the model outputs need to be used at some point. BatchBALD’s runtime complexity is also stated incorrectly, see Kirsch 2019.**
>
> The reviewer is correct -- we apologize for the oversight. The runtime complexity should indeed have the number of classes factor and should be $O(|D_U|m^2bC)$ with $c$ being the number of classes. For BatchBALD, we made a typo and conflated the 2 different samplings that BatchBALD is doing when writing down the complexity. The correct complexity should be $O(|D_U|b\cdot C\cdot m\cdot m’)$ where $m’$ is the number of sampled configurations of $y_{1:n-1}$. We have updated the paper with the corrections.

---

> > ### Author Response · Authors · 2020-11-23
> > **Re:AnonReviewer3 part 2**
> >
> > “In general, the paper is not clearly written, with a notation that changes between the main paper and appendix and variables whose meaning is not explained in the text: for example, the proof of Proposition 2, or the definition of HSIC in the appendix, which does not explain from which distributions the variables are sampled specifically.”
> >
> > We have added in the definition of dHSIC in the Appendix as well. As noted already in the appendix, dHSIC is equivalent to HSIC when d=2 which is the case for us. The definition of HSIC in the appendix is presented in terms of any sampling distribution. For our particular setting, the distribution would be the categorical distribution parameterized by the class probabilities output by the neural network for classification.

---

> > > ### Comment · AnonReviewer3 · 2020-11-24
> > > **Response to the Authors**
> > >
> > > Thank you for the detailed comments and updates to the paper. I still find the paper hard to read, but I'll take the updates and replies into account and update my score accordingly.
> > >
> > > > > Moreover, the step from having a (MC-Dropout-based) BNN with sampled output probabilities for each pool sample to computing kernels for different variables is not clear. Is consistent MC dropout used? How are the kernels computed?
> > >
> > > > If by consistent MC dropout, the reviewer means whether we use the same set of dropout masks for the entire pool set, then yes.[...] We then pass this distance matrix through the kernel function to get a kernel matrix of size again .
> > >
> > > > For a batch kernel matrix, we average together the kernel matrices of the individual examples constituting the batch.
> > >
> > > This somewhat important step is not mentioned anywhere in the paper according to my knowledge. Could the authors please add this?
> > >
> > > My remaining serious concern is that the methodology is not made sufficiently clear in the paper (see above), which will make reproduction difficult. The provided code is of good quality overall, but this reviewer strongly believes that a successful implementation should be possible without looking at the code, and not mentioning having to use fixed dropout masks to preserve correlations between samples is not minor omission.

---

> > > > ### Author Response · Authors · 2020-11-24
> > > > **Re:AnonReviewer3**
> > > >
> > > > We thank the reviewer for their quick response. We mention averaging of the kernel matrices in the section "Scaling $\alpha_{ICAL}$ estimation" -- "For regular ICAL, we average the kernel matrices of points in the candidate batch..." and we will emphasize it further.
> > > >
> > > > MC-dropout masks are fixed for different points to get the joint distribution. We will add a mention in the text. We do want to note that this is standard practice and is used by competing methods like BatchBALD as well.

---

> > > > > ### Comment · AnonReviewer3 · 2020-11-24
> > > > > **Response to the Authors**
> > > > >
> > > > > BatchBALD is the only method so far that makes use of consistent MC-dropout according to the reviewer's knowledge. It might be good practice in general, but it might not be standard practice yet.
> > > > >
> > > > > Thank you for making this more explicit in the paper.

---

> > > > > > ### Author Response · Authors · 2020-11-24
> > > > > > **Re:AnonReviewe3**
> > > > > >
> > > > > > We have uploaded a new version having the clarification. We thank the reviewer again for engaging and updating their score post rebuttal!

---

### Official Review · AnonReviewer4 · 2020-10-28
**Interesting idea for active learning: Well motivated and tested.**

**Rating:** 8
**Confidence:** 4

**Review:**

**Summary:** This paper proposes a way to do batch mode model agnostic active learning. In this task, the agent has to query a batch of data points from a set of unlabeled examples for which it will get labels. The paper puts an additional requirement that the algorithm is model-agnostic. The key idea here is to sample a batch of points that provide the most "information" about the remaining unlabeled examples.  Authors argue that this will result in higher performance on the unlabeled examples. The proposed approach called ICAL (Information Condensing Active Learning) uses Hilbert-Schmidt Independence Criterion (HSIC) to measure dependence between a chosen batch and the unlabeled examples. The goal is to pick a batch with a maximum value of HSIC which should intuitively give us a batch which is representative of the unlabeled set. HSIC can be easily estimated unlike other dependence measures such as mutual information. Given a batch size $|B|$, a dataset of unlabeled examples $D_u$  and $m$ samples to estimate HSIC, the ICAL algorithm computes a batch for label acquisition in $O(|D_U|m^2|B|)$ steps, where a greedy strategy is used to search over batch.  Results are presented over MNIST, variants of MNIST and CIFA and show improvements over five previous active learning approaches and a random acquisition baseline.

**Strength:**
1. ICAL makes a useful contribution to the active learning literature which has wide range use. Particularly, experiments are presented on realistic domains, with neural networks, and show gains over a few different baselines.
2. ICAL is model-agnostic which means it can be applied to decision trees, neural networks, complex ensembles, etc.
3. Experiments show that ICAL acquires a more diverse batch for acquisition.

**Weakness:**
1. For HSIC, one has to decide the kernel which may be difficult for some domains.
2. No comparison with BADGE (Ash et al. 2019) is provided.
3. Time complexity of $O(|D_U|m^2|B|)$ seems expensive particularly if $m > 1000$. What is the value of $m$ that one should expect in practice?

**Questions:**
1. For Repeated MNIST task, how do the different baselines compare in terms of the number of times they pick a datapoint and its replica for label acquisition?
2. Is the set $X$ and $Y$ assumed to be countable in Section 3 since summations are used everywhere.

**Writing:**
There are issues with writing in several places. Some are enumerated below:
1. Grammar error on the second line of the third paragraph of intro in "directly focus on 'minimize' the error rate". It should be minimizing.
2. Unexpected full stop after "model's parameters" in the third paragraph of the intro.
3. No $d \theta$ in Section 3 when taking integral in the first paragraph.
4. What is $B_{D_u}$ in the definition of $B^\star$ in the third paragraph of Section 3. Also, how is the training set used in this definition?

---

> ### Author Response · Authors · 2020-11-23
> **Re:AnonReviewer 4**
>
> We appreciate reviewer 4’s constructive and thoughtful comments, and we thank the reviewer for finding our work interesting and useful, and that model-agnostic aspect of the work important. We would like to address the review’s concerns as following:
>
> **For Repeated MNIST task, how do the different baselines compare in terms of the number of times they pick a datapoint and its replica for label acquisition?**
>
> We thank the reviewer for bringing up this interesting question and we have included a new figure in the supplement which plots the number of times different numbers of copies of the same replica appears in the acquired batches. As shown in Appendix Figure 8, our method (as well as BatchBALD, BayesCoreset and Random) successfully avoided acquiring any redundant copies of the same sample, whereas FASS and Max Entropy acquired 3 copies of the same replica in majority of the acquisitions. This proves that batched active learning strategies are indeed better in sample diversity.
>
> **Is the set $X$ and $Y$ assumed to be countable in Section 3 since summations are used everywhere**
>
> While in the paper, we only consider categorical distributions which do have a countable support, in general, our method applies straightforwardly to any distributions (as long as one can obtain samples from it) including distributions with uncountable support. As the paper only deals with distributions of countable support, we indeed made that assumption for Section 3 for ease of presentation.
>
> **No comparison with BADGE (Ash et al. 2019) is provided**
>
> As the major focus of the work is on model agnostic active learning methods, we have not included BADGE which is not a model agnostic method (it requires computing the hallucinated gradient space with respect to the weights in the last layer of the neural network being used)
>
> **Time complexity of $O(|D_U|m^2B)$ seems expensive particularly if $m > 1000$ . What is the value of $m$  that one should expect in practice?**
>
> Ideally the more MC samples are used the better approximation can be computed. In practice we found that for smaller datasets such as MNIST/EMNIST/fMNIST, m=50 suffice the purpose. For CIFAR10 we used 130 and for CIFAR100 which has much more output classes we used 288. We selected the parameters to maintain reasonable runtime and memory consumption, while keeping the number of inference samples large enough for reliable estimation.

---

### Decision · Program_Chairs · 2021-01-07
**Final Decision**

**Decision:**

Reject

**Comment:**

The paper introduces a model-agnostic heuristic for batch active learning.  There was an agreement among the reviewers that it's a good approach to try and report about, but the paper was ultimately rejected after calibration.

There were two concerns raised in the reviews, and the authors are encouraged to address them in a revision:

1) Several reviewers commented on issues with readability, affecting the paper's reproducibility (see reviews for details).  The reviewers would have also liked to see more evidence of empirical robustness to various choices made.

2) For the paper to be compelling, it should either compare with gradient-based approaches like BADGE (Ash et al. 2019) or include experiments with a representation where BADGE can't be applied (to support the model-agnostic distinction the authors are making).  The core motivation is the same, with both approaches trying to explicitly incorporate predictive uncertainty and sample diversity, and it would be interesting to see a comparison.